# Disease-Specific Quality of Life among Patients with Peripheral Artery Disease in Hungary

**DOI:** 10.3390/ijerph20043558

**Published:** 2023-02-17

**Authors:** Lilla Horváth, Imre Boncz, Zsuzsanna Kívés, Gergely Fehér, Noémi Németh, Fanni Luca Kajos, Katalin Biró, Krisztina Fendrik, Katalin Koltai, Gábor Késmárky, Dóra Endrei

**Affiliations:** 1Centre for Occupational Medicine, Medical School, University of Pécs, 7624 Pécs, Hungary; 2Institute for Health Insurance, Faculty of Health Sciences, University of Pécs, 7621 Pécs, Hungary; 3National Laboratory on Human Reproduction, University of Pécs, 7624 Pécs, Hungary; 4Department of Primary Health Care, University of Pécs, 7623 Pécs, Hungary; 51st Department of Internal Medicine, Medical School, Clinical Center, University of Pécs, 7624 Pécs, Hungary

**Keywords:** peripheral artery disease, questionnaire validation, PADQoL, health related quality of life

## Abstract

Peripheral artery disease (PAD) is a progressive atherosclerotic disease significantly impacting functional status and health-related quality of life (HRQoL). This study aimed to investigate HRQoL among PAD patients in Hungary using the validated Hungarian version of the PADQoL questionnaire. Patients with symptomatic PAD were consecutively recruited from the Department of Angiology, Clinical Center, University of Pécs, Hungary. Demographics, risk factors, and comorbidities were registered. Disease severity was measured by Fontaine and WIFI stages. Descriptive statistical analysis, Chi-square test, and non-parametric tests were performed (*p* < 0.05). Overall, 129 patients (mean age 67.6 ± 11.9 years, men 51.9%) participated in our study. The Hungarian PADQoL demonstrated good internal consistency (α range: 0.745–0.910). Factors on intimate and social relationships gave the best (89.15 ± 20.91; 63.17 ± 26.05) and sexual function (28.64 ± 27.42), and limitations in physical functioning (24.68 ± 11.40) the worst scores. PAD had a significant negative impact on the social relationships of patients aged 21–54 years (51.6 ± 25.4). Fontaine stage IV patients experienced significantly lower HRQoL due to fear and uncertainty (46.3 ± 20.9) and limited physical functioning (33.2 ± 24.8). The Hungarian PADQoL identified central aspects of HRQoL. Advanced PAD was found to impact several areas of HRQoL, primarily physical functioning and psycho-social well-being, drawing attention to the importance of early diagnosis and management.

## 1. Introduction

Past decades have seen non-communicable diseases (NCDs) emerging as leading causes of morbidity and mortality worldwide, associated with an increasing disease burden in developing and developed regions alike. As cardiovascular diseases have become the main causes of mortality and morbidity, the prevalence of peripheral artery disease (PAD) has also been rising markedly [1,2,3]. According to most recent estimates, 236.62 million people above 25 years of age were living with PAD in 2015 [4].

Peripheral artery disease is a progressive atherosclerotic disease that results in stenosis or obstruction of the peripheral arteries and is an indicator of generalized atherosclerosis [5,6,7]. The presentation of PAD varies considerably, with many patients remaining asymptomatic for a while or experiencing only mild symptoms. Individuals with early stage PAD either do not experience or frequently under-report claudication symptoms (pain in the lower extremities triggered by walking due to reduced blood supply), and despite the high in-hospital costs associated with advanced stage PAD, the disease often remains undetected and untreated [8,9,10,11,12]. Symptomatic patients may experience claudication, cramping pain in the calves, thighs or buttocks, characteristically related to exercise such as walking or stair climbing, usually subsiding with rest. Further symptoms may also include atypical pain on exertion and ischemic pain at rest. Untreated or advanced stage PAD may lead to tissue loss and amputation [13,14]. Risk factors for PAD include age, race, smoking, hypertension, diabetes, and hyperlipidemia [15,16,17]. PAD is associated with considerable physical and psychosocial disease burden, primarily resulting from impaired functional status and quality of life [18,19,20,21,22] Studies conducted in the U.S. and Hungary have highlighted that functional impairment and pain resulting from lower extremity arterial ischemia leads to reduced quality of life and, consequently, a considerable proportion of PAD patients experience severe symptoms of anxiety and depression. Studies have found that it is mainly impaired the lower extremity functioning that lies at the root of an impaired psycho-emotional state, considerably affecting engagement in social activities and family life [23,24]. Mental health concerns, especially stress, are highly prevalent among PAD patients experiencing new or worsening symptoms. Higher stress levels have been shown to impede successful disease management [25]. Despite advanced endovascular and surgical limb salvage interventions becoming more available, the incidence of PAD-related major and minor amputations has continued to remain alarmingly high, especially, in Eastern European countries, with Hungary continuing to have the highest rates [26,27]. In spite of increasing efforts, PAD has continued to be underdiagnosed and undertreated [8,28,29].

As the past decades have seen a considerable rise in the number of multinational and multicultural research projects, the use of health-related quality of life (HRQoL) instruments in these studies, and consequently, the need to translate and adapt these health status measures for use in other than the source language has also grown rapidly. The increasingly international nature of drug trials has created a demand for translated instruments, primarily to enable a comparison or aggregation of results across different language groups. In terms of health policy, there is a need for indicators that can be used in monitoring the health of populations and for program evaluation. Increasing attention is also being paid to cross-cultural comparisons of effective interventions and preferences for different states of health in order to facilitate the extrapolation of results from effectiveness and cost-effectiveness studies from one country to the other. [30] Besides traditionally used clinical outcome measures for evaluating the impact of a certain medical condition as well as the effects of invasive procedures, recent decades have seen a considerable increase in the application of generic and disease specific HRQoL questionnaires. Although clinical outcome measures in PAD including walking capacity (e.g., treadmill testing), physiological measurements such as the ankle-brachial index (ABI), patency tests of revascularized segments, or amputation-free survival provide an adequate and clear picture of the patients’ objective clinical status, HRQoL measures, patient reported outcomes tools have been proven to add valuable information about the actual daily functioning of PAD patients. Furthermore, disease specific HRQoLs that have been developed and constructed to assess changes in the quality of life in subpopulations of patients suffering from a particular medical condition can assess the social and emotional consequences of living with a particular disease or help compare HRQoL prior to and after particular therapeutic interventions. It has been suggested that both generic and disease-specific questionnaires are required in quality of life research [31,32]. Disease-specific questionnaires have been found to be able to discriminate more precisely between major versus more minor changes in disease severity, subsequent to interventions in PAD [33].

As Hungary has seen an alarmingly high level of major amputations related to PAD, our aim was to provide a further validated tool that can assess the subjective disease burden of patients with PAD experience. Peripheral artery disease quality of life (PADQoL) is a validated questionnaire, designed to assess disease-specific physical, psychosocial, and emotional effects of PAD [34]. We aimed to assess the quality of life among patients suffering from various stages of PAD using the Hungarian version of the PADQoL questionnaire we have previously validated.

## 2. Materials and Methods

The type of research conducted was a quantitative cross-sectional study. The study was carried out at the Department of Angiology, 1st Department of Internal Medicine, Clinical Center, University of Pécs, Hungary. The study was conducted between March 2020 and November 2021. Patients were consecutively enrolled through purposive sampling. Informed consent was obtained from the study participants prior to completing the survey.

### 2.1. Data Collection Instrument: PADQoL

PADQoL is a disease-specific questionnaire developed by Treat-Jacobson et al. with the aim to provide a further validated tool for the assessment and evaluation of the impact of PAD on HRQoL from the aspect of the subjective burden of disease. The 38-item questionnaire investigates five factors: social relationships and interactions (nine items), self-concept and feelings (seven items), symptoms and limitations in physical functioning (eight items), fear and uncertainty (four items), positive adaptation (seven items), and contains three individual items relating to job, sexual function, and intimate relationships. Response options were scored on a 6-point Likert-type scale ranging from 1 (strongly agree) to 6 (strongly disagree). Two of the individual items (sexual function, intimate relationships) and the summed scores on Factor 5 were reverse coded. The developers’ aim was to construct an instrument that, aside from limitations in physical functioning, focused more on the subjectively perceived social and emotional burden of PAD and its effects on well-being and quality of life. PADQoL is different from other existing PAD-specific health-related quality of life instruments (e.g., the WELCH (Walking Estimated-Limitation Calculated by History questionnaire that has been used to assess walking limitation in a PAD patient population in Hungary) [31] in that it is able to assess the relative differences for the individual patient, and may thus provide effective guidance to clinicians in choosing treatment strategies and help identify patients who may require additional therapies including psychotherapy or social support [34].

### 2.2. The Cross-Cultural Adaptation Process

The linguistic validation of self-administered health-related quality of life (HRQoL) questionnaires requires a unique method to ensure both linguistic and cultural equivalence between the original source (SL) and target language (TL) versions.

The linguistic validation of the PADQoL was carried out according to the international guidelines [35,36,37] as follows: two certified translators translated the questionnaire including the introduction, items, and response options from English into Hungarian. The so called ‘informed’ translator was specialized in medical and health-sciences translation and interpreting, and the other translator, the ‘uninformed’ translator, had a post-graduate certificate in economics and social sciences translation and interpreting. The ‘informed’ translator received information about the aim of the questionnaire and the concepts it was designed to investigate. As PADQoL is a self-administered health status measure, the two forward translators were recommended to focus more on maintaining a conceptual dynamic rather than formal equivalence. A third certified translator reviewed the two forward translations (T1 and T2) with the two forward translators, discussed inconsistencies, and agreed upon a consensus version (T3). Working from the consensus version (T3), totally blind to the original, two back translators with a non-medical background created two back translations. Back translators were native speakers of English, had been living in Hungary for more than 30 years, and were fluent in Hungarian. The back translations did not reveal gross inconsistencies or conceptual errors. A consensus team consisting of the forward translators, the back translators, the third consensus translator, two internist-angiology specialist physicians, a linguist, and an applied linguist reviewed and discussed all translations. The team reached a consensus on discrepancies and created the pre-final Hungarian version of PADQoL, which then underwent pilot-testing through cognitive interview sessions involving 30 patients. The pilot patient population consisted of native speakers of Hungarian who adequately represented the target population socio-demographically and clinically in terms of age, sex, education, diagnosis, and previous interventions including revascularization and comorbidities. None of the Hungarian items proved difficult to understand or were inappropriate at a conceptual level. Minor changes were necessary in the wording of the items’ linguistic aspects and results of the pilot testing have been previously published [38,39].

### 2.3. The Study Population

One hundred twenty-nine patients from in- and outpatient care were involved in the study. Participants (males, females) with established diagnosis of PAD (Fontaine II–IV), aged 18 and above who agreed to participate in the research and were able to fill in the questionnaires independently were included. There was no upper age limit, patients who were unable to complete the questionnaire due to their mental or physical condition were excluded. Fontaine stage I patients were also excluded from our study as the instrument used for measuring disease-specific quality of life was developed and designed for surveying patients living with symptomatic PAD. Patients who had lower leg ulcer (Fontaine IV) not only due to PAD but also due to chronic venous insufficiency were also excluded.

Informed consent was obtained from the study participants prior to completing the survey. Participation was voluntary. All patients completed a paper-based PADQoL instrument during a face-to-face interview independently. Demographics included were gender, age, and educational achievement. Risk factors and comorbidities included body mass index (BMI), smoking, diabetes, hypertension, dyslipidemia, coronary heart disease, stroke/TIA, carotid artery narrowing, chronic kidney disease, chronic obstructive pulmonary disease (COPD), musculoskeletal disorders, and previous revascularization.

Physical examination, resting ankle-brachial index (ABI) ≤ 0.9 was used as an indicator of disease. In patients with borderline ABI (0.90–1.00), further diagnostic tests were carried out as normal ABI does not definitely rule out the diagnosis of LEAD; in such cases, post-exercise ABI and Doppler ultrasonography were performed. In the case of a high ABI (>1.40) related to medial calcification, toe pressure and toe-brachial index (TBI) were measured [13]. The severity of patients was evaluated with the Fontaine and the WIFI classification systems. The Fontaine Classification system assesses the clinical presentation of patients, that is, symptoms ranging from asymptomatic PAD to the presentation of necrosis and/or gangrene in the lower extremities, and four stages are established indicating disease severity [40]. The WIFI classification system covers the three most essential parameters that may indicate the risk of a future amputation: wound, ischemia, and the presence of foot infection. This classification system attributes a 4-grade scale to each parameter, ranging from 0 to 3, where 0 represents absent, 1 mild, 2 moderate, and 3 severe risk [41,42].

### 2.4. Statistical Analysis

Descriptive statistical methods were used, and the mean ± SD (standard deviation), absolute and relative frequency were calculated. To examine the association between categorical variables Chi-square test, in the case of continuous variables, the normality test (Shapiro–Wilk test) was performed on data presenting non-normal distribution, and non-parametric tests were carried out (Mann–Whitney U test, Independent samples Kruskal-Wallis test with pairwise comparisons) at the 95% confidence interval (CI), (*p* < 0.05). Data analysis was performed using SPSS (version 22.0, IBM).

### 2.5. Ethical Approval

The study was approved by the Regional and Institutional Research Ethics Committee of the Clinical Center of the University of Pécs (8106-PTE 2019).

## 3. Results

### 3.1. Sociodemographic Characteristics of Patients

A total of 129 patients completed our questionnaire survey, 62 women (48.1%) and 67 men (51.9%). The mean age was 67.6 years, SD 11.9 years, with a median of 70 years. The youngest participant was 21, the oldest was 89 years old. There was no significant difference between the sexes in the different age groups (*p* = 0.950).

Regarding educational achievement, there were significantly more women among the participants with primary education only (28.4%; *p* = 0.027). There was no significant difference among the age groups with regard to educational background (*p* = 0.244).

With regard to smoking, 32 participants (24.8%) never smoked, 73 (56.6%) were past smokers, and 24 patients (18.6%) were current smokers. There was no significant difference between the sexes in terms of smoking (*p* = 0.211), although we found a higher percentage of women (22.4%) compared to men (14.5%) among the current smokers. Compared to educational achievement, no significant difference was found regarding smoking (*p* = 0.605), although there were more current smokers (28.0%) among those participants who only had an elementary education background compared to those who completed secondary (18.1%) or higher education (9.5%).

The mean BMI in our study population was 27.8 kg/m^2^, SD: 6 kg/m^2^; median: 27.3 kg/m^2^. The average BMI among males was 28.5 kg/m^2^ (SD:5.6 kg/m^2^) and among women 27.0 kg/m^2^ (SD:6.3 kg/m^2^); we found no significant difference between the sexes (*p* = 0.174). We found no significant differences in the BMI when comparing age groups (*p* = 0.492), sexes (*p* = 0.302), or educational background (*p* = 0.785). Comparing the BMI categories with smoking yielded no significant differences either (*p* = 0.239).

Considering the risk factors, 38% (49/129) were overweight, 29.4% (38/129) of participants were obese, and 18.6% (24/129) patients were current smokers. More than half of the patients (55.0% (71/129) of patients) had a diagnosis of diabetes, the majority of patients (88.4% (114/129)) had hypertension, and 89.1% (115/129) had dyslipidemia. Regarding comorbidities, 65.9% (85/129) of our participants had coronary heart disease (CHD), 50.4% (65/129) had carotid artery stenosis, 27.1% (35/129) suffered from chronic kidney disease, 17.8% (23/129) had had a stroke or transitory ischemic attack (TIA), 16.3% (21/129) had COPD, and 64.3% (83 patients) suffered from a musculoskeletal disorder. More than half of the study population, 56.6%, had undergone previous revascularization (bypass or endovascular therapy) (Table 1).

### 3.2. Severity of PAD

Less than one third of patients had Fontaine stage IIa (37/129; 28.7%), or IIb PAD (38/129; 29.5%), 14 participants (10.9%) suffered from stage III disease and every third patient had severe, stage IV PAD (40/129; 30.1%) (Table 2). WIFI values, indicating amputation risk within a year, were calculated for patients with Fontaine stage III and stage IV disease. Out of the 54 patients who met this criterion, according to the WIFI categories, 3.7% (2 patients) had very low, 22.2% (12 patients) had low, 18.5% (10 patients) had moderate, and more than half, 55.6% (30 patients), had a high risk for possibly having to undergo an amputation in the near future (Table 3). The number of volunteers in the Fontaine III group (N = 14) and WIFI groups ‘very low’ (N = 2) and ‘high’ (N = 30) were outliers.

### 3.3. PADQoL Questionnaire Survey Results

Table 4 provides a summary of the mean, standard deviation, possible range, and minimum and maximum scores for this sample. Item 24 ’Intimate relationships’ and Factor 1 ’Social relationships and interactions’ gave the highest (i.e., the best) scores. The worst (lowest) scores were found in the case of Item 23’Sexual function’, and the Factors ‘Symptoms and limitations in physical functioning’ and ’Fear and uncertainty’.

Factor weights exceeded the minimum value of 0.25 in all variables. Factor weights were very similar to those of the original questionnaire (not shown in table).

The reliability of the PADQoL subscales was assessed through calculating the Cronbach’s alpha values ranging between 0.745 and 0.910 in our patient population. The internal consistency of the majority of the factors was excellent. Internal consistency values were the highest in Factor 1 and the lowest in Factor 5, similar to those of the original measure. The Cronbach’s alpha values of the validated questionnaire were very similar to those of the original questionnaire, which proves that the translated Hungarian tool is reliable and measures selected aspects of HRQoL adequately (Table 5.).

Sex and educational achievement showed no significant correlation with any of the factors. Regarding age groups, patients aged 21–54 had significantly lower average scores in the Factor ’Social relationships and interactions’ (F1) than those aged 65–74 years. Participants aged 75 years and above had significantly worse scores in Factors ’Positive adaptation’ (F5) and ’Sexual function’ (F7) compared to those between 21–54 and 65–67 years. The factor measuring work performance, ’Job’ (F6), showed that compared to the younger age spectrum (21–54-year old participants), those aged 65–74 and ≤75 years had significantly higher scores.

Item 24 ‘Intimate relationships’ showed high scores in all age-groups, in both sexes and was not significantly affected by educational background either.

Item 23 measuring sexual function showed that the worst mean values significantly deteriorated above age 64 compared to the younger age groups. The Factor ‘Symptoms and limitations in physical functioning’ also indicated a worse quality of life in our study population but showed no significant differences with regard to age, sex, or educational background. Patients aged 21–54 years reported their social relationships and interactions to be more negatively impacted by PAD compared to those aged between 65 and 74 years. Positive adaptation was found to significantly deteriorate with age above 65 years. Item 21, measuring the impact of PAD on patients’ work, revealed a worse quality of life in this respect among the younger patients. Item 24, assessing ‘Intimate relationships’, gave the highest mean values (Table 6).

Upon comparing Fontaine stages with the PADQoL subscales, we found that the mean values in Factors F1 and F3 decreased with disease severity. Factor F2 also showed a decrease with advancing PAD with Fontaine stages III and IV patients reporting a significantly lower quality of life and increasing levels of inner fears and uncertainty compared to patients with stage IIa disease (Table 7). Item 23, measuring sexual function, showed the impact of PAD on this segment of life early on and further decline. Factor F5 and Items 21 and 24 did not show significant change with advancing disease.

## 4. Discussion

We intended to investigate the HRQoL among patients suffering from PAD using a disease-specific quality of life instrument we had previously validated and cross-culturally adapted for use in Hungary. As the global burden of PAD has also increased over the past decades in developed and developing regions of the world, the early detection of the disease, especially among asymptomatic patients or those with atypical symptoms, is of crucial importance, as PAD has remained underdiagnosed and untreated [28,29].

PAD is associated with significant morbidity and mortality from cardio- and cerebrovascular diseases and patients have an equal risk of suffering a future stroke or MI as patients with coronary artery disease [43,44]. PAD is associated with considerable physical and psychosocial disease burden, primarily resulting from declining physical/functional status and deterioration in the quality of life [18,19,45].

Besides a healthy diet, physical activity and controlled exercise training, smoking cessation is one of the most important elements of CV risk factor management. Unfortunately, Hungary has not shown the desired levels of a reduction in smoking among the general population, which is also reflected by the fact that more than half of our participants were past, and nearly 20% were current smokers, despite having been diagnosed with PAD [46]. Lower levels of education were associated with higher rates of smoking.

The fact that more than half of our patient population had diabetes, and a significant majority had hypertension and dyslipidemia underline the importance of early diagnosis and adequate pharmacological therapy including up-to-date lipid control (statin, ezetimibe) and adequate antiplatelet therapy [13]. As PAD is a sign of generalized atherosclerosis, it was not unusual to find that more than half of our patients had CHD, or carotid artery narrowing. Previous stroke/TIA were also present in nearly one-fifth of our patients. Although our sample size was quite small, our findings emphasize the importance of a need for more complex screening for CVD events, especially among patients with asymptomatic disease [47].

Although more than half of the study population had undergone previous revascularization (bypass or endovascular therapy), Hungary and other Eastern European countries are in a 10-year delay with regard to these types of interventions being prioritized as the first-line option [48,49]. The current sad situation is also highlighted by the fact that every third of our patients had severe, stage IV PAD, and more than half were at high risk of a future amputation.

The PADQoL questionnaire survey results clearly demonstrated the impact of PAD on sexual and physical function, together with increased levels of anxiety and fear among patients suffering from the disease becoming more expressed with age. Our findings also underline the extent to which PAD impacts upon social life, interfering with friendships and aspects of social life function. Apparently, younger patients felt more severe limitations in terms of the social relationships caused by PAD as with age, and due to other comorbidities, older people tend to become used to being limited in their social lives. Intimate relationships were the least affected area of life, irrespective of age or disease severity, the majority of our patients reported as having been satisfied with this aspect of their lives, which underlines the importance of the support coming from close relationships with family and friends in coping with a progressive medical condition, especially as positive adaptation showed a considerable decrease in older patients in our study, and worse perceived health status has been shown to be associated with deteriorating mental health [50,51].

The PADQoL factor measuring work performance showed that compared to the younger age spectrum, participants above 65 years scored significantly higher, emphasizing the fact that PAD progresses with age, and consequently, has a more severe impact on the ability to engage in work-related activities.

Comparing the PADQoL factors with Fontaine stages underlined the fact that patients with more advanced PAD are increasingly hindered in keeping contact with friends and relations and in engaging in social activities mainly due to physical pain and the resulting limitations in movement. PAD has a strong impact on the patients’ self-concept and self-worth, and significantly affects psychological well-being. Our survey also found a considerable decrease in this respect with advancing disease, with Fontaine stages III and IV patients reporting significantly lower quality of life and increasing levels of inner fears and uncertainty compared to patients with stage IIa disease. The above clearly demonstrate that PAD has a considerable impact upon multiple aspects of psychological and social life, aside from significantly deteriorating the patients’ physical health, as has been revealed by several previous studies [34,52,53,54]. The number of volunteers in the Fontaine III group and WIFI groups ‘very low’ and ‘high’ were outliers, which might be a limitation of our study. Due to this reason, we provided subgroup analysis only according to Fontaine stages (Kruskal–Wallis test) and not for the WIFI categories.

Finally, our study had some limitations. It was a cross-sectional, observational study in nature including more than 120 patients. The sample was not representative of the general PAD population. The fact that all patients were recruited from one regional clinical center may have resulted in some degree of selection bias. The above-mentioned limitations may have influenced our findings. Additionally, follow-up with regard to the effectiveness of medication therapy or revascularization procedures was not carried out.

## 5. Conclusions

Aside from validating and adapting a new HRQoL questionnaire for use in Hungary, our cross-sectional study measured the HRQoL among patients living with different stages of PAD. Our results revealed that PADQoL can be an effective tool in assessing the quality of life in the Hungarian patient population and could also be used for monitoring the change subsequent to pharmacotherapy or surgical interventions, thereby enabling the comparison of quality of life gained with international data. As our study revealed significant associations between more advanced stages of PAD (Fontaine III and IV) and HRQoL mainly in the areas of social life and mobility markedly impacting emotional health as well as highlighting the pivotal importance of early diagnosis and the adequate management of PAD patients.

## Figures and Tables

**Table 1 ijerph-20-03558-t001:** Sociodemographic data, risk factors, and comorbidities of the study population. Legend: * Asterisked data indicate statistically significant results.

Variables	Total n (%)	Males n (%)	Females n (%)	*p*-Values ^a^
Age groups				
21–54 years	18 (14.0)	8 (12.9)	10 (14.9)	0.950
55–64 years	13 (10.1)	7 (11.3)	6 (9.0)	
65–74 years	59 (45.7)	29 (46.8)	30 (44.8)	
75 years and above	39 (30.2)	18 (29.0)	21 (31.3)	
Educational background				
Less than 8 years of primary school/primary school certificate	25 (19.4)	6 (9.7)	19 (28.4)	0.027 *
Secondary school certificate	83 (64.3)	45 (72.6)	38 (56.7)	
Higher education degree	21 (16.9)	11 (17.7)	10 (14.9)	
Risk factors, Comorbidities				
Smoking				
Never	32 (24.8)	13 (21.0)	19 (28.4)	0.211
Past smoker	73 (56.6)	40 (64.5)	33 (49.3)	
Current smoker	24 (18.6)	9 (14.5)	15 (22.4)	
BMI				
Underweight (≤18.49 kg/m^2^)	7 (5.4)	2 (3.2)	5 (7.5)	0.412
Normal (18.50–24.99 kg/m^2^)	35 (27.1)	14 (22.6)	21 (31.3)	
Overweight (25.00–29.99 kg/m^2^)	49 (38.0)	28 (45.2)	21 (31.3)	
Obesity grade I.(30.00–34.99 kg/m^2^)	23 (17.8)	10 (16.1)	13 (19.4)	
Obesity grade II. (≥35.00 kg/m^2^)	15 (11.6)	8 (12.9)	7 (10.4)	
Diabetes				
No	58 (45.0)	26 (41.9)	32 (47.8)	0.506
Yes	71 (55.0)	36 (58.1)	35 (52.2)	
High blood pressure				
No	15 (11.6)	8 (12.9)	7 (10.4)	0.664
Yes	114 (88.4)	54 (87.1)	60 (89.6)	
Dyslipidemia				
No	14 (10.9)	6 (9.7)	8 (11.9)	0.680
Yes	115 (89.1)	56 (90.3)	59 (88.1)	
Coronary heart disease				
No	44 (34.1)	21 (33.9)	23 (34.3)	0.956
Yes	85 (65.9)	41 (66.1)	44 (65.7)	
Stroke/TIA				
No	106 (82.2)	52 (83.9)	54 (80.6)	0.627
Yes	23 (17.8)	10 (16.1)	13 (19.4)	
Carotid artery narrowing				
No	64 (49.6)	33 (53.2)	31 (46.3)	0.430
Yes	65 (50.4)	29 (46.8)	36 (53.7)	
Chronic kidney disease				
No	94 (72.9)	47 (75.8)	47 (70.1)	0.470
Yes	35 (27.1)	15 (24.2)	20 (29.9)	
COPD				
No	108 (83.7)	50 (80.6)	58 (86.6)	0.363
Yes	21 (16.3)	12 (19.4)	9 (13.4)	
Musculoskeletal disorder				
No	46 (35.7)	27 (43.5)	19 (28.4)	0.072
Yes	83 (64.3)	35 (56.5)	48 (71.6)	
Previous revascularization				
No	56 (43.4)	26 (41.9)	30 (44.8)	0.745
Yes	73 (56.6)	36 (58.1)	37 (55.2)	

^a^ Chi-square test; * *p* < 0.05.

**Table 2 ijerph-20-03558-t002:** Fontaine stages in the study population (n = 129).

Fontaine Stages	No. of Patients	Distribution (%)
IIa	37	28.7
IIb	38	29.5
III	14	10.9
IV	40	31.0
Total	129	100

**Table 3 ijerph-20-03558-t003:** WIFI categories in the study population (n = 54).

WIFI Category	No. of Patients	Distribution (%)
Very low	2	3.7
Low	12	22.2
Moderate	10	18.5
High	30	55.6
Total	54	100

**Table 4 ijerph-20-03558-t004:** The PADQoL questionnaire survey results.

PADQOL Factors	Mean (SD)	Possible Range	Minimum	Maximum
Factor 1. Social relationships and interactions, summed score	37.43 (11.72)	9–54	12	54
Factor 1. Social relationships and interactions, transformed score	63.17 (26.05)	0–100%	7	100
Factor 2. Self-concept and feelings, summed score	26.24 (9.58)	7–42	7	42
Factor 2. Self-concept and feelings, transformed score	54.97 (27.38)	0–100%	0	100
Factor 3. Symptoms and limitations in physical functioning summed score	24.68 (11.40)	8–48	9	48
Factor 3. Symptoms and limitations in physical functioning transformed score	41.37 (28.50)	0–100%	3	100
Factor 4. Fear and uncertainty summed score	14.88 (5.03)	4–24	6	24
Factor 4. Fear and uncertainty transformed score	54.42 (25.18)	0–100%	10	100
Factor 5. Positive adaptation summed score	26.09 (5.07)	7–42	9	35
Factor 5. Positive adaptation transformed score	54.55 (14.50)	0–100%	6	80
Item 21. Job summed score	3.87 (1.95)	1–6	1	6
Item 21. Job transformed score	57.36 (39.00)	0–100%	0	100
Item 23. Sexual function summed score	2.40 (1.32)	1–6	1	6
Item 23. Sexual function transformed score	28.64 (27.42)	0–100%	0	100
Item 24. Intimate relationships summed score	5.46 (1.04)	1–6	1	6
Item 24, Intimate relationships transformed score	89.15 (20.91)	0–100%	0	100

**Table 5 ijerph-20-03558-t005:** The PADQoL internal consistency of the pre-test and final Hungarian versions compared to the original version.

PADQOL Factors	Cronbach’s Alpha n = 129	Cronbach’s Alpha n = 297 [34]
Factor 1. Social relationships and interactions	0.91	0.92
Factor 2. Self-concept and feelings	0.91	0.89
Factor 3. Symptoms and limitations in physical functioning	0.90	0.88
Factor 4. Fear and uncertainty	0.78	0.80
Factor 5. Positive adaptation	0.74	0.73

**Table 6 ijerph-20-03558-t006:** Correlation between the mean values of the PADQoL factors with the sociodemographic data. Legend: * Asterisked data indicate statistically significant results.

Variables	N	F1. Social Relationships and Interactions	F2. Self-Concept and Feelings	F3. Symptoms and Limitations in Physical Functioning	F4. Fear and Uncertainty	F5. Positive Adaptation	Item 21. Job	Item 23. Sexual Function	Item 24. Intimate Relationships
All patients	129	63.2 ± 26.1	55.0 ± 27.4	41.4 ± 28.5	54.4 ± 25.2	54.6 ± 14.5	57.4 ± 39.0	28.6 ± 27.4	89.2 ± 21.0
Sexes									
males	62	63.5 ± 24.8	55.1 ± 27.3	39.8 ± 28.6	52.1 ± 26.2	56.5 ± 13.1	56.7 ± 40.0	28.0 ± 26.4	91.3 ± 16.0
females	67	62.8 ± 27.7	54.8 ± 27.6	43.5 ± 28.4	56.6 ± 24.2	52.7 ± 15.5	57.9 ± 38.3	27.8 ± 26.5	87.2 ± 24.5
*p*-values ^b^		0.940	0.979	0.445	0.285	0.201	0.996	0.889	0.505
Ages									
21–54 years	18	51.6 ± 26.4	48.7 ± 27.7	31.1 ± 28.3	51.7 ± 26.4	61.9 ± 9.0	26.7 ± 40.9	45.6 ± 35.5	93.3 ± 13.7
55–64 years	13	70.7 ± 20.2	62.8 ± 20.3	43.0 ± 29.0	47.6 ± 24.2	55.8 ± 12.6	35.4 ± 40.9	52.3 ± 23.8	86.1 ± 28.7
65–74 years	59	68.4 ± 25.6	59.0 ± 28.1	47.6 ± 27.3	58.7 ± 24.2	55.7 ± 14.2	66.4 ± 34.9	26.1 ± 24.1	87.1 ± 23.1
>75 years	39	58.0 ± 26.3	49.0 ± 27.2	37.8 ± 29.2	51.4 ± 26.1	49.0 ± 15.9	65.1 ± 34.5	14.3 ± 13.7	91.2 ± 17.0
*p*-value^s a^		0.040 *	0.177	0.085	0.354	0.011 *	<0.001 *	<0.001 *	0.657
Educational background									
Less than 8 years of primary school/primary school certificate	25	61.9 ± 25.4	53.6 ± 30.0	37.9 ± 25.6	52.0 ± 24.6	52.2 ± 15.0	52.0 ± 38.7	28.0 ± 24.4	94.4 ± 10.8
Secondary school certificate	83	62.1 ± 25.9	54.9 ± 26.5	41.9 ± 29.6	54.7 ± 26.2	54.2 ± 14.5	56.6 ± 39.4	27.9 ± 26.1	88.2 ± 21.4
College/university	21	63.2 ± 28.3	56.9 ± 28.5	45.5 ± 28.0	55.9 ± 22.4	58.6 ± 13.6	69.7 ± 37.5	27.6 ± 31.3	86.7 ± 27.0
*p*-value ^a^		0.932	0.942	0.700	0.850	0.280	0.273	0.874	0.537

^a^ Kruskal–Wallis test, ^b^ Mann–Whitney U test.

**Table 7 ijerph-20-03558-t007:** The correlation between Fontaine stages and PADQoL factors. Legend: * Asterisked data indicate statistically significant results.

Fontaine Stages	N	F1. Social Relationships and Interactions	F2. Self-Concept and Feelings	F3. Symptoms and Limitations in Physical Functioning	F4. Fear and Uncertainty	F5. Positive Adaptation	Item 21. Job	Item 23. Sexual Function	Item 24. Intimate Relationships
IIa	37	77.9 ± 18.6	65.7 ± 25.2	57.7 ± 27.7	64.5 ± 25.8	54.4 ± 16.0	68.11 ± 38.7	30.8 ± 26.4	87.5 ± 21.2
IIb	38	59.3 ± 28.9	52.8 ± 30.0	38.9 ± 29.4	56.0 ± 27.2	53.4 ± 15.9	51.5 ± 40.9	33.1 ± 28.0	92.6 ± 13.4
III	14	58.1 ± 17.2	57.3 ± 20.3	30.8 ± 21.5	46.0 ± 20.0	58.9 ± 13.2	50.0 ± 37.4	28.5 ± 29.0	97.1 ± 7.2
IV	40	55.0 ± 26.7	46.2 ± 26.1	33.2 ± 24.8	46.3 ± 20.9	54.0 ± 12.0	55.5 ± 37.2	20.0 ± 23.0	84.5 ± 27.7
*p*-value ^a^		<0.001 *	0.020 *	<0.001 *	0.011 *	0.720	0.182	0.101	0.245

^a^ Kruskal–Wallis test.

## Data Availability

The dataset supporting the conclusions of this article is available upon request to the corresponding author.

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
