# Peer review of "Disease-Specific Quality of Life among Patients with Peripheral Artery Disease in Hungary"

_ijerph, 2023, doi:10.3390/ijerph20043558_

Round 1
Reviewer 1 Report
Title: it is understood that the psychometric properties make up the list of characteristics and necessary for the validation of an instrument, thus suggesting removing this term.
Abstract
The objective does not match the title, it implies that the psychometric properties of the already validated instrument will be analyzed. This should be reviewed, and reanalyzed (title and objective),
You should give preference to the method of describing the population over the statistical analysis that will be addressed in the text, so this item should be redone.
In the result to present the main findings, this was not done.
The conclusion must respond to the objective
Introduction
I missed the introduction to problematize the importance of validation, the citation of other validations of the same instrument, the clinical and scientific implications of validation.
The objective presented differs from that presented in the abstract.
Methods
How was the randomization done?
On line 113, it refers to a specific PAD questionnaire, but does not name it, I suggest indicating the name and, if it has validation for Hungarian, also make this reference.
The other validation tests and analyzes were not conducted: pre-test, triangulation, expert evaluation, justify.
I understand that the comparison was made with HRqol, so reference should be made to its validation for the language in question.
For validation between the questionnaires, it is suggested to use a confidence test, eg. kappa.
Results
It is not necessary to describe what is already exposed in the tables
The number of volunteers at different stages of PAD was statistically the same¿
The number of volunteers in the different stages of the wifi categories was statistically the same¿
Present the result of the suggested analyzes
Discussion
Present the discussion of the results of the suggested analyzes
Conclusion
Redo or revise aim
Author Response
Response to Reviewer 1.
We would hereby like to thank you for your valuable insights and comments
- The text of the article has been reviewed and corrected by a certified translator-reviewer.
- Title: it is understood that the psychometric properties make up the list of characteristics and necessary for the validation of an instrument, thus suggesting removing this term.
The title has been changed.
Abstract
- The objective does not match the title, it implies that the psychometric properties of the already validated instrument will be analyzed. This should be reviewed, and reanalyzed (title and objective),
Title and objectives have been modified.
- You should give preference to the method of describing the population over the statistical analysis that will be addressed in the text, so this item should be redone.
Abstract has been modified.
- In the result to present the main findings, this was not done.
Abstract has been modified to present the main findings.
- The conclusion must respond to the objective
Abstract has been modified and conclusion was revised.
Introduction
- I missed the introduction to problematize the importance of validation, the citation of other validations of the same instrument, the clinical and scientific implications of validation.
A section has been added on the importance of the validation of self-reported health status measures and their clinical and scientific implications. Although there exist other language versions of PADQoL, (Korean, Canadian) as indicated on the developers’ website, no validation studies into other languages have been found.
- The objective presented differs from that presented in the abstract.
The study objective has been modified and synchronized in abstract and in text.
Methods
- How was the randomization done?
Patient recruitment method has been described and changed to purposeful sampling, as follows:
„Patients with Fonatine II-IV.stage symptomatic PAD referred to the Department of Angiology, Clinical Centre, University of Pécs for outpatient or inpatient care were consecutively enrolled via purposeful sampling. Patients aged 18 and above who agreed to participate in the research and were able to fill in the questionnaires independently were included. There was no upper age limit, patients who were unable to complete the questionnaire due to their mental or physical condition were excluded . Fontaine Stage I patients have no symptoms. Therefore, they were excluded from our study as the PAD QoL instrument used for measuring disease-specific quality of life was developed and designed for surveying patients living with symptomatic PAD.”
- On line 113, it refers to a specific PAD questionnaire, but does not name it, I suggest indicating the name and, if it has validation for Hungarian, also make this reference.
Reference has been added to a paper discussing the results of the Short-tERm cIlostazol eFFicacy and quality of life (SHERIFF) study conducted by Farkas et.al in which authors used the Hungarian version of the WELCH (Walking Estimated-Limitation Calculated by History) questionnaire. We have found no prior validation study of the WELCH questionnaire into Hungarian.
- The other validation tests and analyzes were not conducted: pre-test, triangulation, expert evaluation, justify.
Sub-chapter 2.2 shortly describes the validation process. However, the validation process (cross-cultural adaptation process discusses the forward-, back-translation, consensus translation, expert commitee and pilot-testing procedures) was already published in a separate paper (References 30,38,39)
- I understand that the comparison was made with HRqol, so reference should be made to its validation for the language in question.
References 30,38,39 discuss the validation procedure of PADQoL questionnaire in details including linguistic validation and pilot testing.
- For validation between the questionnaires, it is suggested to use a confidence test, eg. kappa.
The validation of PADQol (including different statistical tests) into Hungarian has already been performed and its results have been published, see references 30,38,39.
Results
- It is not necessary to describe what is already exposed in the tables
Results section has been shortened.
- The number of volunteers at different stages of PAD was statistically the same?
The number of patients was slightly outlier in different stages of PAD.
- The number of volunteers in the different stages of the wifi categories was statistically the same?
The number of patients was slightly outlier in different wifi categories.
- Present the result of the suggested analyzes
We performed detailed subgroup analysis only for Fontaine stages (results presented in Table 7.) Kruskal-Wallis test was used for checking statistical significance which includes the estimation of differences in the number of patients in the independent samples (stages).
Discussion
- Present the discussion of the results of the suggested analyzes
We mentioned this issue at the end of discussion.
Conclusion
- Redo or revise aim
The aim of the study has been modified.
Reviewer 2 Report
More specific comments are as follows:
1. Currently, the abstract contains 264 words and its structure does not follow the requirements set by the IJERPH. there is no need to list the whole spectrum of statistical methods used in the abstract.
2. The first paragraph of the Introduction section (lines 38-46) is not relevant for the study aim.
3. The authors should describe the process of patient enrollment in more detail rather than stating that it was "random".
4. Although the authors reported the use of a broad spectrum of statistical tests, it is not clear what test was used in every particular case. In this regard, the manuscript would benefit from inclusion of the name of statistical test used in each table.
5. The discussion section is too short and does not allow an adequate interpretation of the study findings. Also, the study limitations have to be presented in Discussion section, not Conclusions. The authors need to present strategies that were applied to mitigate them.
Author Response
Response to Reviewer 2.
We would hereby like to thank you for your valuable insights and comments
1, The text of the article has been reviewed and corrected by a certified translator-reviewer.
- Currently, the abstract contains 264 words and its structure does not follow the requirements set by the IJERPH. there is no need to list the whole spectrum of statistical methods used in the abstract.
The abstract has been shortened, and modified according to your suggestions.
- The first paragraph of the Introduction section (lines 38-46) is not relevant for the study aim.
The first paragraph has been shortened.
- The authors should describe the process of patient enrollment in more detail rather than stating that it was "random".
Patient recruitment has been changed to purposive sampling, and has been described in more detail.
“The study was conducted at one University Clinic: the Department of Angiology, Clinical Center, University of Pécs. Due to reallocation of medical staff during the Covid-19 epidemic, patients suffering from PAD had limited access to outpatient care which resulted in a considerably smaller number of patients available for our questionnaire survey.”
Further exclusion criteria were added in the text as follows:
„There was no upper age limit, patients who were unable to complete the questionnaire due to their mental or physical condition were excluded. Fontaine Stage I patients were also excluded from our study as the instrument used for measuring disease-specific quality of life was developed and designed for surveying patients living with symptomatic PAD. .Patients who had lower leg ulcer (Fontaine IV) not only due to PAD, but also due to chronic venous insufficiency were excluded as well.”
- Although the authors reported the use of a broad spectrum of statistical tests, it is not clear what test was used in every particular case. In this regard, the manuscript would benefit from inclusion of the name of statistical test used in each table.
Names of statistical tests used have been added to each table.
- The discussion section is too short and does not allow an adequate interpretation of the study findings. Also, the study limitations have to be presented in Discussion section, not Conclusions. The authors need to present strategies that were applied to mitigate them
We modified the discussion section. Study limitations section has been moved to the Discussion section.
Reviewer 3 Report
The current manuscript titled: "Validation and Psychometric Properties of the Hungarian Version of the Disease-specific Health-related Quality of Life Instrument PADQoL" represents an important analysis of evolving field of Internal Medicine, Vascular Medicine and Cardiology.
In my opinion, these are the adjustments which should be made to increase the value of your manuscript:
1. Line 25-26: please change the sentence more clearly “24 (18.6%) were current smokers Dyslipidaemia (89.1%),”.
2. Line 32: please remove the dots after “III” and “IV”.
3. In Introduction chapter: taking into account the fact that the quality of life is also reflected in the psycho-emotional state, please describe the relationship between anxiety, depression and stress with PAD.
4. The small number of patients is not entirely justified, despite the fact that the study was conducted in 2 University Clinics with a vascular profile in period of 19 months and the Authors described that the patients were consecutively enrolled. Explain these discrepancies, it may be necessary to add clearer exclusion criteria, or it would be good to increase the number of patients included in the study.
5. In the Methodology section, please add a detailed description of the PAD diagnosis used criteria.
6. Please, explain in the text why patients stage I Fontaine were excluded from the study.
7. According to which classification were the patients divided into age categories?
8. For reliable statistical processing and validation of this scale, the following statistical analyzes must be added: for model calibration - Hosmer-Lemeshow goodness-of-fit χ2 statistics, for discriminative ability of the model – c statistic, AUC, ROC.
9. Please move the Limitations section before the Conclusions section.
10. The manuscript contains some punctuation errors, please revise the text.
11. Adapt the references according to the Journal requirements.
Author Response
Response to Reviewer3.
We would hereby like to thank you for your valuable insights and comments
- The text of the article has been reviewed and corrected by a certified translator-reviewer.
- Line 25-26: please change the sentence more clearly “24 (18.6%) were current smokers Dyslipidaemia (89.1%),”.
Punctuation error has been corrected.
- Line 32: please remove the dots after “III” and “IV”.
Dots indicated have been removed throughout the text.
- In Introduction chapter: taking into account the fact that the quality of life is also reflected in the psycho-emotional state, please describe the relationship between anxiety, depression and stress with PAD.
The relationship between anxiety, depression and stress with PAD has been discussed in more detail in the introduction.
- The small number of patients is not entirely justified, despite the fact that the study was conducted in 2 University Clinics with a vascular profile in period of 19 months and the Authors described that the patients were consecutively enrolled. Explain these discrepancies, it may be necessary to add clearer exclusion criteria, or it would be good to increase the number of patients included in the study.
The study was conducted at one University Clinic: the Department of Angiology, Clinical Center, University of Pécs. Due to reallocation of medical staff during the Covid-19 epidemic, patients suffering from PAD had limited access to outpatient care which resulted in a considerably smaller number of patients available for our questionnaire survey.
Further exclusion criteria were added in the text as follows:
There was no upper age limit, patients who were unable to complete the questionnaire due to their mental or physical condition were excluded. Fontaine Stage I patients were also excluded from our study as the instrument used for measuring disease-specific quality of life was developed and designed for surveying patients living with symptomatic PAD. Patients who had lower leg ulcer (Fontaine IV) not only due to PAD, but also, due to chronic venous insufficiency were excluded as well.
- In the Methodology section, please add a detailed description of the PAD diagnosis used criteria.
Physical examination, resting ankle-brachial index (ABI) ≤0.9 was used for PAD diagnosis. In patients with borderline ABI (0.90–1.00) further diagnostic tests were carried out as normal ABI does not definitely rule out the diagnosis of PAD. In such cases post-exercise ABI and doppler ultrasonography were performed. In case of a high ABI (>1.40) related to medial calcification, toe pressure, toe-brachial index (TBI) were measured.
- Please, explain in the text why patients stage I Fontaine were excluded from the study.
Fontaine Stage I patients have no symptoms. Therefore, they were excluded from our study as the PAD QoL instrument used for measuring disease-specific quality of life was developed and designed for surveying patients living with symptomatic PAD.
- According to which classification were the patients divided into age categories?
We divided patients into age categories according to the epidemiological and clinical characteristics of peripheral artery disease.
- For reliable statistical processing and validation of this scale, the following statistical analyzes must be added: for model calibration - Hosmer-Lemeshow goodness-of-fit χ2 statistics, for discriminative ability of the model – c statistic, AUC, ROC.
Sub-chapter 2.2 shortly describes the validation process we have previously performed (cross-cultural adaptation process the forward-, back-translation, consensus translation, expert committee and pilot-testing procedures) and the results of which have been published in a separate papers (References 38,39,40)
The validation of PADQol (including different statistical tests) into Hungarian has already been performed and its results have been published, see references 38-40.
- Please move the Limitations section before the Conclusions section.
The limitations section has been moved as requested.
- The manuscript contains some punctuation errors, please revise the text.
The text of the article has been reviewed and corrected by a certified translator-reviewer.
- Adapt the references according to the Journal requirements.
References have been corrected according to Journal requirements.
Round 2
Reviewer 2 Report
Well done!!
Reviewer 3 Report
I agree with the changes made, which significantly improve the quality of the manuscript.